# From Havana to Cádiz in the Imaginary of Women Writers of the Last Decades

María del Mar López-Cabrales [1,*] and Inmaculada Rodríguez-Cunill [2]

1    Department of Languages, Literatures and Cultures, Colorado State University, Clark Building C-104, Fort Collins, CO 80523, USA
2    Department of Painting, University of Seville, Laraña 3, 41003 Seville, Spain; cunill@us.es
*    Correspondence: maria.lopez-cabrales@colostate.edu

**Abstract:** In this essay, we intend to demonstrate how the cities of Havana and Cádiz became mutable literary subjects that accompany the female characters of the narratives of female writers of the past decades from Havana (Anna Lidia Vega Serova, Ena Lucía Portela, and Mylene Fernández Pintado) and Cádiz (Ana Rossetti and Pilar Paz Pasamar). The ironic and delusional visions of a ruined life due to the special period, economic crisis, and political xenophobia in Cádiz will be illustrated by Cuban-Spanish mapping of the analyzed authors' works. Our hypothesis stems from the idea that there is a clear relation between the representation of the city and political, cultural, and patriarchal transgression that is quoted in these texts (Bataille), which relates to the experience of scarcity/poverty lived on both sides of the Atlantic Ocean. Our bibliographic search has focused on the literary expression of the experience of these cities from the point of view of female writers and protagonists. We concluded with a universal understanding of the experience of the space marked by literature and the gaze of women.

**Keywords:** Havana; Cádiz; contemporary female writers; city; women; patriarchy; special period; economic crisis; scarcity

## 1. Introduction

Havana and Cádiz have always been sister port cities, almost twins. The relationship between Havana and Cádiz is relevant for several reasons:

1.    Havana and Cádiz have a long economic and cultural exchange dating back to colonial times. Then, Cádiz was the main port of entry of goods from the Americas to Spain, and Havana became the center of Spanish trade in the Caribbean.
2.    In terms of cultural heritage, Old Havana has been declared a World Heritage Site by UNESCO, highlighting the common cultural heritage of both cities. On the other hand, Cádiz is recognized as a "Good of Cultural Interest" in the Spanish legislation, including material and intangible cultural heritage.
3.    Economically, both cities have strong ties in the tourism and maritime industries. In addition, many cruise ships and ferry services operate between the two cities, providing, in principle, employment opportunities for local communities.
4.    The relationship between Havana and Cádiz is also crucial regarding diplomatic relations between Spain and Cuba. Spain has been one of Cuba's historical allies and trading partners, and the relationship between the two cities symbolizes this partnership.

In Antonio Burgos' famous poem, "Habaneras de Cádiz", he documents precisely that. Both cities flourished in the same centuries (from XVI to XVIII) and are architecturally similar: they have bridges and fortifications built to protect from the English pirate Drake in Cádiz and to protect from the British siege in Havana in 1762. The malecón of Havana, if elongated, would resemble the Campo del Sur in Cádiz. The large cathedrals and historical

centers of both cities are aesthetically very similar. In this essay, we intend to demonstrate how the cities of Havana and Cádiz became mutable literary subjects that accompany the female characters of the narratives of female writers of the past decades from Havana (Anna Lidia Vega Serova, Ena Lucía Portela, and Mylene Fernández Pintado) and Cádiz (Ana Rossetti y Pilar Paz Pasamar). It will be essential to have as a background the relationship between sexuality and architecture, as we will see below (Agrest et al. 1996; Squier 1984; Bloomer 1992).

The ironic and delusional visions of a ruined life because of the special period[1] in Havana and because of the economic crisis and political xenophobia in Cádiz will be illustrated by Cuban-Spanish mapping of the analyzed authors' works. Our hypothesis stems from the idea that there is a clear relation between the representation of the city and political, cultural, and patriarchal transgression that is quoted in these texts (Bataille 2010), which relates to the experience of scarcity/poverty lived on both sides of the Atlantic Ocean.

Bataille's representation of the city and political, cultural, and patriarchal transgression is characterized by a deep-seated sense of ambivalence and paradox. On the one hand, he celebrates the city as a site of intense creativity, vitality, and diversity, where individuals can break free from the constraints of tradition and custom and explore new forms of experience and expression. However, on the other hand, Bataille is acutely aware of the darker, more sinister aspects of urban life, such as poverty, crime, and social inequality. He also recognizes that the city is a site of intense political struggle, where different groups and individuals compete for power and influence (Bataille 2010; Mambrol 2017).

In Bataille's view, these conflicting impulses are intimately linked to the concept of transgression, which he defines as breaking social norms and taboos. Bataille argues that transgression is a necessary and inevitable part of human life, as it allows individuals to challenge existing power structures and create new forms of social organization. However, Bataille acknowledges that transgression can be dangerous and destructive, leading to violence, suffering, and chaos. He, therefore, advocates for a radical and ethical transgression that challenges dominant power structures while respecting all human beings' inherent dignity and worth.

Among these ideas about architecture and sex (mainly since we focus on women writers who establish the imagery of similar cities), some of the following theses that were brought to the table with the book "The Sex of Architecture" (Agrest et al. 1996) should be highlighted:

1. Architecture is a gendered field: architects and architectural spaces have historically been dominated by men, and the profession has propagated gender stereotypes through design.
2. Feminist perspectives can improve architecture: feminist interventions in architecture can reveal and challenge gender biases present in the field, leading to more equitable and inclusive designs. For example, our analysis of literary texts related to the cities of Cádiz and Havana can provide insights into implementing visions of cities that the patriarchal gaze has hidden.
3. Gendered spaces influence behavior: architecture can reinforce or challenge gender roles, creating spaces that enable or constrain certain behaviors and interactions. Our female literati will show us those sights that come out of the mainstream.
4. The Role of Power in Architecture: power dynamics, privilege, and cultural norms shape the practice of architecture and influence who has access to design and decision-making processes.

In this sense, both Bataille's thesis and those concerning gender bias in architecture seem to bear the same idea of power from different viewpoints. In sum, the intersections between gender, sexuality, and architecture challenge traditional notions of the field and advocate for more inclusive and equitable approaches to design. Moreover, the views of the women writers who are the protagonists of our research indicate this openness and overcoming of mainstream thinking.

Suppose we have these ideas as a guide. In that case, it will be possible to understand the analysis we have made of the literary texts that are the protagonists of the imagery of the cities of Havana and Cádiz throughout the following pages. These works could form part of what Doris Sommer titled *Foundational Fictions* (Sommer 1991) but would better fit in with what Anderson called "imagined Community" (Anderson 2006), given that they represent the Cuban and Andalusian communities.

## 2. Imagery in Practice on Both Sides of the Atlantic Ocean

In the case of Cuba, with a capital city that is almost hostile towards its disabled inhabitants, in the middle of the special period—as portrayed in *Yocandra in the Paradise of Nada (La Nada Cotidiana)* by Zoé Valdés (1997), *One Hundred Bottles on a Wall (Cien Botellas en una Pared)* by Ena Lucía Portela (2003) and *Night Round (Noche de Ronda)* by Anna Lidia Vega Serova (2003)—Havana is an exuberant city, full of life, protective and almost a confidant of its residents in Attended Prayers by Milene Fernández Pintado (2003). The Andalusian space will be analyzed using the texts of Ana Rossetti and Pilar Paz Pasamar as a place full of light and diversity, where they speak about the bay, the salt flats, the estuaries, and the sea; finally, we added a story by Rodríguez-Cunill (2021) that reinforces our comments about these places. But Cádiz has also become a conflictive area due to the immigration of people from the African continent.

Havana and Cádiz are located in areas that in our lifetimes do not only represent localization but become cybernetic cities (Boyer 1996), entities, carriers of their own text and with semiotics that their population have to decode (Mangieri 1994) to survive within them: billboards, storefront displays, music, neon lights, traveling salesmen, video clips, graffiti, advertisements in English, and phones are instances where the translation is necessary to understand the urban Imaginarium and to, in this way, defeat the tensions and norms of the urban framework. Concretely, Havana represents a singular example, as instead of the habitual billboard advertisement that incites consumption, one sees billboards with political slogans, such as "Cuba cannot be sold", "Cuba goes", "Revolution or death", "Socialism or death! Motherland or death! We will come out victorious!", "Here we do not want masters", and "Until victory, always", among others.

In the case of Cádiz, the city's graffiti is mostly in protest against political corruption, and, despite the effort of authorities to paint over or take them down, one can constantly find comical or poetic phrases about any and everything in the most depressed neighborhoods of the city, namely in: Santa María, La Viña, El Pópulo, Guillén Moreno or Puntales. Some examples of this are the anti-war graffiti "–Are they the enemy?/–Well, we surrender. Why?/–Because this makes NO SENSE", or the call for rebellion: "Official reveal yourself, the nation-state strangles you" or the image of a smoking piece of excrement subtitled: "Don't complain if when you vote your hands smell like shit" that as a black and white flier covered the walls of Cádiz before the general elections of 2011.

In addition to mass unemployment, the changes in the city of Cádiz in recent decades are characterized by industrial de-structuring, the (re)emergence of large pockets of social exclusion, and the city's exploitation towards a model of tourist consumption and privatization of public space. All this has led to the reactivation of conflicts among the population. As we shall see, given the city as the literary subject of our writers, their texts are parallel voices that, in a certain way, react to these fundamental changes in the city.

There is a direct relationship between, on the one hand, the dismantling of industry and, on the other, the consequences of this dismantling for the majority of the city's inhabitants, in terms of the rise of the tourist model and the increasing privatization of public spaces.

On the other hand, political power has joined the development of urban policies conceived on the basis of the need for a significant event as the main machine for the city's transformation. Although this fact is not new (livestock fairs, Olympic games, etc.), we want to emphasize that it introduces the need for marketing to create an image of the city to attract investment. Moreover, this brand was invented before the change in the town took

place, making the previous picture an indispensable condition for urban transformation. In the game of imagery, the city as a literature subject is based on very different concerns.

In the last decades, the city of Cádiz has incessantly tried to include itself in the competition of the city market and the current new neoliberal city model. All this has involved the construction of different city brands ("Cádiz, the city that smiles" or "Cádiz 2012: Constitutional City") based on the archetypal construction of spaces and a showcase city, where to camouflage social conflicts with a supposed cosmetic cohesion that continues to attract visitors.

Geographically, the fact that Havana is located on an island and that the city of Cádiz, due to its enclave, has insular characteristics leads us to find more coincidences in the imagery we are examining. Due to the city's urban distribution, its growth is unfeasible. The connections of Cádiz by land are currently three: the road that joins San Fernando with Cádiz, the Carranza bridge that connects Puerto Real with the city, and a third entrance exit, the Constitution of 1812 bridge. The last one is known as "Second Bridge" or "La Pepa's Bridge", as "La Pepa" is how this legal text is popularly known. The "Second Bridge" pharaonic works have been part of the imagery of the people of Cádiz from 2008 to 2015 (the date of its inauguration) and beyond. In addition, during that period, references have been made to the "Second Bridge" in the lyrics of carnival songs, which are renewed annually. As we can see, the name of the bridge coincides with one of the brand images of the city that the public authorities have promoted.

The brand "Cádiz, the city that smiles" has also elicited a comprehensive creative response from the authors of carnival compositions. From irony, Soto and Vargas wrote:

> "What a slogan the publicists have come up with now ( . . . ) Cádiz, the city that smiles . . . Cádiz is dying of laughter as it sees drugs crossing the Strait of Gibraltar and corpses stranded on its shores. The unemployment rate is highest in Spain, and Cádiz is laughing. The people of Cádiz are so funny that we would invent our new slogan: 'Cádiz does not smile.... Cádiz dies laughing, Cádiz is laughing its ass off. ( . . . ) Well, if Cádiz is smiling, it's not funny to me at all'". ([Soto and Vargas 2007](#))

Until the 1980s, the city's main economic activity was the naval and aeronautical industry, with thousands of workers employed in these activities. But in recent decades, a major restructuring has left a minimal part of this industry in private hands. In this process, thousands of jobs were lost, which has led to a high rate of emigration of the young population. This has led to a decrease in the number of young people and therefore in the number of inhabitants in the city and, on the other hand, to the aging of the population. Another consequence of this process is the increased evictions and cuts in basic supplies such as water and electricity. Neighborhoods such as El Cerro del Moro, San Mateo, or Guillén Moreno are the most affected by evictions due to lack of income. These evictions have occurred mainly due to non-payment of rent and non-payment of mortgages. In Cádiz, as in many other cities in Spain, access to housing has become a severe problem that intensifies segregation and inequalities in urban space.

An image that became typical are the lines in specific centers where food is distributed to families with fewer resources, as well as the crowds in the city's soup kitchens. Many people of Cádiz have faced problems with primary needs such as food. The similarities with the special period in Cuba are clear.

On the other hand, the underground economy, which already played an important role before the so-called market crisis, is still a common sight on the city's streets: the sale of fish, second-hand stalls, the clandestine lottery: women (almost all of them) selling lottery tickets or shares are characteristic elements of the city's neighborhoods.

In addition, as a fundamental characteristic of this process, the city has turned to tourism, which has increased and diversified the services offered to visitors, such as hotels, restaurants, and tour guide companies. In this sense, the daily arrival of cruise ships to the port of Cádiz has been intensified, but this does not mean an increase in the city's economy, but rather an increase in the number of visitors because the services for tourists are already

guaranteed in the port of Cádiz, on the ships, so that the cruise passengers invade the city with their presence. Still, they do not bring any benefit to most of the inhabitants.

### 3. Havana and Cádiz, Literary Subjects

In 1836, the poem "Al partir" by Tula, Gertrudis Gómez de Avellaneda, after her departure from Cuba, observes the Cuban woman's unparalleled love for her birthplace. This loving bond for places starts from a female voice, which has not been usual in the history of literature, where the male voice has been associated with the power of the territory:

> "Pearl of the sea! Star of the West!/Beautiful Cuba! Your bright sky/covers the nights with its opaque/veil as it covers my pain and my sorrow/[...] Farewell, my homeland, beloved Eden!/wherever I am, its fury employs me,/your sweet name praises the ear! [...]".[2]

Yocandra (in *Yocandra in the Paradise of Nada*) explains her routine during the special period in the following:

> "For two years I have been doing the same thing every day: bike from my house to my office, punch in, sit at my desk, read a few foreign magazines that keep coming in two or three months late, and think about cobwebs [...]. I come home, it's dark out. I start cooking at three, but the gas comes and goes, so before I know it it's eight in the afternoon. At that time if I can eat, I can consider myself a realized woman [...]. When I finish eating [if I can] I clean the house and before going to bed I read something, or I watch some movie if by then the electricity is back on". (Valdés 1997)

What in Gómez de Avellaneda is a look from the loss, in the protagonist of *Yocandra in the Paradise of Nada* is an observation within the same city, from the experience of the routine that gives scarcity during the special period. In a way, they are two losses: the last economic one and the first one of the surrounding universe. These two voices, so distant in time, have common elements. The loss is transcendent from the absence, the scarcity and precariousness, the emptiness of the routine, and the slow despair that evicts women from their spaces. The city's vision is alien to voices outside the mainstream, as we would see both in Bataille and in studies on the sex of architecture. And, despite that, the discomfort created by Yocandra's pedaling in *Yocandra in the Paradise of Nada* becomes, in this novel, a liberating urban experience in Havana after the special period:

> "And I pedal with the energy of a cyclist and a premiered lover; I almost am able to become the French girl from the movie, and then my fantasies become lyrical pornography, while people cross the narrow streets, flanked by houses without doorways, by shirtless men wearing flip flops, their torsos coming out of their pants or shorts, that cat call women with tissues and curls, who wear tight shirts over spandex under which compress their bodies, which play hide and seek with cleavage and hems. [...]. They carry buckets of water with resignment as their daily gesture. But bitterness is a passive noun, and so it must leave way for many other things. For a walk, a laugh, or scream, to impose at home there is the street: life". (Valdés 1997)

Zeta, the protagonist in *One Hundred Bottles on a Wall* (2002), describes her experience similarly:

> "In those days, my economic situation wasn't going too well. In all honesty, it wasn't going at all: it had been paralyzed. I can't tell if we lived on the edge of collapse or if we were already in it. I had lost my job as a redactor for that dark neighborhood about agricultural issues [...]. The day before (and the one before that, and the one before that ... ) I had gone to bed drunk. [...] I felt [...] very afflicted by the concert of marks on my stomach, because if anything depresses us fatties, it's problems about food". (Portela 2003)

In Portela's text, plagued with humor—or jokes—and irony, Zeta[3] speaks without stopping the flow of consciousness stemming from the mansion, the Vedado in which she was born as a true architectural gem, "a monument to extravagance, and a prodigy for scraps, patches, and sewing an eclectic Frankenstein by the fashion standards of 1926 and a ruin by the fashion standards of the years that followed" and she compares her "own room" with the one belonging to Virginia Woolf:

"[...] I would love to see Mrs. Woolf writing Mrs. Dalloway between the cackling of the chickens, the thunderous barks of the megatherium, trying to consume the electricity debt collector, or the hutia that is so good, the dominoes on top of the table [...], the grunts of the pig that runs around terrified when he is to be showered with a hose, to see if they can get rid of its stink, the war of the decibels [and a long etcetera] [...]". (Portela 2003; Hayden 1987)

In *Night Round* (2003) by Anna Lidia Vega Serova, the city is presented as a desert where food and water are scarce[4].

"Bunny Banana never imagined that a city full of desolate streets could be the Sahara Desert. She always thought that beyond her bedroom door [...] began the paths of real life, filled with lights, music, and happy people; the same paths towards happiness 'I was so wrong!'".

Nevertheless, Havana is portrayed, in *Otras Plegarias Atendidas* (*Attended Prayers*) (2003) by Milene Fernández Pintado, as a place where problems and difficulties exist but also as a loved and yearned-for distant place[5].

Batman, the protagonist's lover, leaves for Miami and explains the falsehood about the lack of time with which exiled people live to convince themselves that they did the right thing and, in this way, avoid yearning for their homeland:

"It is not true, there only one moment in each day during which you are left with nothing to distract yourself from remembering. [In Havana] there is madness, misery, claustrophobia, desperation, but there is no solitude. Life is shared to an extent where the national sin is promiscuity. The sun of Havana shines to bring for there to be any corner in which anything can be hidden". (Fernández Pintado 2003)

When Batman leaves for Miami, the protagonist does not capitulate, nor does her city. Havana lives and does not collapse, despite the abandonment it suffers at the hands of its population, and although it yearns for their return to feel complete again, it has its own resources with which to fill this whole:

"[...] Havana is not a woman that is brought down by infidelity. She will remake herself like she has on so many occasions. Perhaps it is the most lascivious lover and abandons the world. But in her, there are lubricated nights, and many panting mornings so as to dedicate canvas prints belonging to Penélope". (Fernández Pintado 2003)

Fernández Pintado, in the same way Tula did two centuries prior, expresses her love for Havana:

"You still have not left the city, full of blackouts, homilies, and exaggerations, but you have already forgiven these flaws; you have turned the present to nostalgia, you have filtered the present and kept the memories that have always been good. A place that is one thousand times over maledictus loses its oppressive contours and becomes a dreamed space to which you go for protection in every moment of abandonment which you suffer, and it has you, clinging mutually to that lacerating ownership". (Fernández Pintado 2003)

As we can see, in the extracts of texts by Cuban women writers that evidence the sensation of loss, always associated with the temporal dimension, we find a spirit of a certain sadness that combines, paradoxically, with the lustrous experience of the tropics. The contradictions that Bataille shows in this tour provide evidence of how the Havana

imagery develops. Still, our next step is to discover how this view of the city elaborated by women writers is unraveled.

Cádiz and the province that surrounds it have been immersed in a constant economic and social crisis starting from the so-called "industrial conversion" (or the closing of the shipyards) realized by Felipe González's socialist government after winning elections from 1982, beginning in 1984 together with Carlos Solchaga.[6] Cádiz is one of the places with the highest unemployment rates and juvenile exile due to the lack of jobs across all of Spain. Nevertheless, in Ana Rossetti's story "Maud's Kingdom" the description made by Maud of the beach and her surroundings is one of a liberating space where one feels safe and vindicated, despite the feelings of exclusion she goes through by her male friends that consider her to be incapable of knowing more than them (López-Cabrales 2000). Maud promises to survive on her island[7] so that they can understand her and so no one dares to disagree with her. The bay and the beach are two open and luminous spaces:

> "The bay if full of boats and treasures[...]/I am like the bay. [...]/Wherever I go, I am on my island. Inside is all that I own. [...]/I look at the sleeping sea of the bay, and I know that deeper in there are cargo ships carrying wonders like a fabulous kingdom. [...] The bay also is the unknown kingdom belonging to Maud. But I do not feel sadness any longer. I make a circle in the sand, and I situate myself in its center, looking out towards the bay".

Although Rossetti is not a writer characteristic for setting her texts in her place of birth, in her poem "Anonymous memories that I had no choice but forgetting in a pianist's van" from her book of poems, *The Dalliances of Erato* (1980), there is a clear allusion to the salt flats, the estuaries of San Fernando and the mural from Pompeii of Venus Afrodite Anadiomene, going out to sea as a metaphor for the whiteness of the salt flats. Not in vain, Cádiz is named the "savory clarity":

> "Minute estuaries. Salt diapers/sunken in your hands. [...] [...]
>
> The red salt in the estuary jumped, purely.
>
> The white Anadiomene of September,
>
> nude of her pink tunics,
>
> crepitates, luminous, under the sun".

From the prose, the story "La piedra de Inma la Magnánima" ("The stone of Inma the Magnanimous") (Rodríguez-Cunill 2021) leads us, from a point of view similar to Rossetti's to an implication in the reality of survival in today's Cádiz. What the protagonist finds in the rocky area of the beach of La Caleta are what she understands to be Phoenician archaeological remains. The whole story moves towards the need to name after her (and recover afterward) a carved stone she finds on that beach (where many natural stones have unique names, and there is even a map of them). In this search for the title, the protagonist (the story is written in the first person) has to negotiate with characters in the city who live off the black market, who traffic in archaeological remains whose findings are not communicated to the institutions responsible for the Historical Heritage. In this clash between the desire for transcendence linked to a city and the process of finding the traces of a previous civilization, the actual town, and the feminine gaze on it, are described. These are not terrible situations but an emotional transformation linked to the feeling of the city. It is not strange that a reflection by Fernando Quiñones on what it is to love a city precedes the story:

> "TO LOVE a city... what is it to love a city? Wouldn't it be to love ourselves in it, everything we were and remember? This may be true, but it is a truth that does not go alone, since in that city, we also love what we did not get to live, what preceded us. And we also have -not infrequently- the case of the late lover, who arrived, saw, and was hooked by the charms of the city that was not his but that is going to be". (Rodríguez-Cunill 2021)

The idea that Cádiz is a space of the confluence of civilizations is present, and the gaze on it is not dispersed but deepens its perspective in specific elements (in this case, a carved stone) that give meaning to the whole city.

In some of the stories written by Pilar Paz Pasamar, the province of Cádiz is presented as a meeting point of Saharan and North African cultures with occidental cultures. For example, the Phoenician women Sidón and Tiro, that would arrive on the Andalusian coasts between the years 100 and 500 A.D., are represented in two of her stories, "The Sea was in Front" and "The Lady of Cádiz". Saharan women are representative of migrant travelers that try to flourish in other lands by risking everything. They die or survive crossing the desert and Alboran Ocean and are referenced in "The Boot for the Right Foot". Here, Paz Pasamar creates Juli and Andres, a pair of socially unfavored citizens of Cádiz that live poorly, trying to profit from the commercialization and sale of illicit drugs, as well as the sale of objects taken from Saharan bodies that are left stranded on the Andalusian coast, after their sterile attempt to cross the strait.

The protagonist in "The Boot for the Right Foot" is Falémé, a name given to her by Juli, her lover and "protector"—her pimp—born in Cádiz, as he never learns the Senegalese woman's name. Falémé, named after a river in western Africa that runs through Senegal and Mali, is a beautiful prostitute who lives braiding her long hair. Looking away from the sea until she finds, in Juli's right boot, the DNA of her murdered Senegalese lover, she takes justice into her own hands and takes revenge for her lover as well as the thousands of undocumented people that cross the desert and the strait. After killing Juli, "she ordered that Juli's body be put in his own boat, taken to sea, and dumped with so many others" (López-Cabrales 2012).[8]

Another text from Paz Pasamar about the meeting of cultures in the Gibraltar Strait happens in "The Ticket" and defends Moroccan women. The text starts with a situational and temporary description frame:

> "[...] Over the strait's waters, the migratory birds head towards northern Africa. [...] An overflow Cádiz, basking in its own light, ceases transit. Its skin is made of the cross paths of different races, bunched together, confused as the same, like rivers that fade into the ocean. The walk of its history is very long and ancient". (López-Cabrales 2012)

The plot of "The Ticket" takes place in the port of Algeciras, which faces the clarity of the port of Cádiz, presenting itself as a sad and decaying place, a port that holds witness to the racking of so many citizens of both sides of the strait:

> "The city, like all coastal cities, emanates the peculiar stench of its port and its ocean, showing prints of smoke and saltpeter. Maritime cities awaken, tired, and the walls of their buildings, like the lime on the walls of the mountain towns, add color to the white. The continuous strokes of smoke and shadow augment a darkness that not even the morning light decreases. The clarity is unable to break off the supposed capes of nocturnality". (López-Cabrales 2012)

In said port, before departing, the protagonist finds herself with a three-year-old Moroccan girl, abandoned by her parents, holding a sign written in three languages: "S'apelle Fátima. Trois ans. Por toi"./– "He name Fátima. Zree years. Four you"./–"Se llama Fátima. Tres años. Quédatela" (López-Cabrales 2012).

The honey-colored eyes of the Moroccan girl—future—against the tired eyes of the occidental woman—past—mix like the waters of the Atlantic Ocean and the Mediterranean Sea in the Alboran Sea. The woman removes her veil—her dark glasses—to read the sign that the girl is holding. Nevertheless, she puts the veil back on—and, following her husband, leaves the girl behind, unable to break her apathy and submission, deliberately blind.

## 4. Conclusions

In our approach to the texts of women writers from both sides of the Atlantic Ocean, we wanted to analyze and profile some cities and cultures through the anguish and joys

experienced by their people, particularly by their women. The texts mentioned above paint places based on their beauty and effervescence. They often go back to eras far removed in time but describe a present thanks to their archaeological remains (especially in the case of Cádiz). In this sense, they fully follow a tradition. Still, the time jump is so significant (from Roman or Phoenician times to the present of the Cádiz area) that several nuances have arisen.

First, it gives the impression that this distant past constitutes a handle to build another image of the city. But, of course, this has happened before since geographical and historical conditions can make a city in a country stand out (even though its foundation is not recent) as a city far removed from what is the mainstream of thought and culture in that country. We have noted this in past research, for example, regarding Nagasaki and the fact that it is the least Japanese city in Japan[9] (Cabeza-Lainez and Rodríguez-Cunill 2019).

Bataille's theses seem to be put more into practice when our women writers focus less on transcendence through the beauty of the past of ancient civilizations and instead attach their misery, hardship, and conflicts to their plots, further showing that these cities are written from the experience and characters of women. This reading of the world from the perspective of the traditionally invisible half of the population offers the most genuine aspect of our article. Working on the margins, from a post-structuralist universe, we advance further in constructing the reality ahead of us. These are complex realities, paradoxical as Bataille celebrated, and in the contemporaneity of scarcity and precariousness (when women are poorer and more precarious, now and traditionally).

The commitment—and sacrifice—to the life of the cities that speak to us joins many other constructions of cities made from the margins, from the voices of minorities trying to survive in a fabric that is not the same as the one in which they were born: in short the secret histories of cities, which shape it beyond the mainstream (Mehta 2017). Despite the fact that we find in our women writers allusions to distant civilizations, they are now allusions to the foreign family, to the lost well-being of years gone by. Contemporaneity marks the conflict—and the humor—of the narratives that have occupied us. Havana and Cádiz are mythical and distant cities. In these propitious spaces, the writings of women authors imagine what lies beyond a disturbed and oppressive "reality", increasing our universe of cities that Rama (1984) called lettered to, now more than ever, hybrid (García Canclini 1990, 1993, 1995).

Therefore, by comparing our twin cities, these writers have taken a step forward without forgetting the resources used by other writers. The human species advances (and deteriorates), and these writers have echoed this. But, as always, we must remember that cities constitute literary subjects, each with its own particularities. In the field of Spanish-American literature, this is what the Mexican writer Elena Garro did in 1963 with the streets of Ixtepec in *Recollections of Things to Come* (*Recuerdos del porvenir*) (five years before García Márquez's Macondo was capitalized by history as a literary subject), the town of Comala, created in *Pedro Páramo* by Juan Rulfo, would acquire a fundamental consistency for the story; the narrative cycle of the city of Santa María would culminate in *The Shipyard* (*El astillero*) by Juan Carlos Onetti. But in universal literature, the city has not ceased to be a transcript of the psychological realities of the protagonists of novels, stories, or poetry[10] because it is through space (with all its geographical and cultural peculiarities) that we constitute and place ourselves as literary beings. The city thus becomes a space of individual and collective creativity, but it is also the product of the same, with its fatigue, sacrifices, concealments, and transgressions (Rodríguez-Cunill 2013).

**Author Contributions:** Conceptualization, M.d.M.L.-C.; methodology, M.d.M.L.-C.; software, I.R.-C.; formal analysis, M.d.M.L.-C. and I.R.-C.; investigation, M.d.M.L.-C. and I.R.-C.; resources, M.d.M.L.-C. and I.R.-C.; writing—original draft preparation, M.d.M.L.-C., and I.R.-C.; writing—review and editing, I.R.-C. and M.d.M.L.-C.; supervision, M.d.M.L.-C. and I.R.-C.; project administration, I.R.-C.; funding acquisition, M.d.M.L.-C. and I.R.-C. All authors have read and agreed to the published version of the manuscript.

**Funding:** This research received no external funding.

**Institutional Review Board Statement:** Not applicable.

**Informed Consent Statement:** Not applicable.

**Data Availability Statement:** Not applicable.

**Acknowledgments:** Author appreciates the generosity of Jonathan Carlyon, Chair of the Department of Languages, Literatures and Cultures of the Colorado State University.

**Conflicts of Interest:** The authors declare no conflict of interest.

## Notes

[1] "The special period in times of peace was an appellative that the Cuban government gave to the crisis that the island suffered after the collapse of the socialist camp, the disintegration of the Soviet Union, and the Soviet decision to terminate all the help and sale of petrol to Cuba [...] During this crisis the US government increased the global blocking of Cuba and approved the Torricelli and Helms-Burton laws. In 1989, Fidel Castro alerted the Cuban people of the collapse of the socialist camp, and in 1990 a plan was created to face the special period in a time of peace due to the resistance of the Cuban population and the concept of 'war of all' relate to the necessary means taken to overcome the global blocking, the systemic attacks and even the hypothetical military intervention on the island" (López-Cabrales 2007b). The special period and the survival tactics created by women, reference Holgado Fernández (2000). In the Mexican context, reference Masolo (1992), and for a more general context reference, Folguera (1987).

[2] In the same way, in the contemporary narratives of female Cuban writers, it is evident that this unconditional love towards a city where, despite the vicissitudes, is possible, even happiness. For example, Cuquita in Zoe Valdes' *I Gave You All I Had (Te di mi vida entera)* (1997) said the following about Havana: "That's the thing about Havana; the more you walk, the more you want to walk. You never get bored, the cities become more beautiful because, on each occasion, it saves a different adventure for you. A seduction that melts one's heart. Although it is falling apart, even if it were to die of disappointment, Havana will always be Havana" (Valdés 1997, p. 73).

[3] "[...] humor and irony are resistance strategies because they serve to reconfigure one's view of transgressive female characters that refuse to capitulate in moments of crisis" (López-Cabrales 2007b). Also, reference the works of Linda Hutcheon about the political connotations of irony in a contemporary world (Hutcheon 1994).

[4] In one of the apartments described in this novel, there is no furniture; instead, small signboards that read 'bed', 'table', 'chair', or 'foreign.' The characters, like in a play, have to act like they are sitting at the table, sleeping on the bed, or looking at the painting when in reality, these are not there (during the special period, people would sell all of their belongings for dollars and use these to buy their necessities, since the national banks were empty and their supply cards failed to provide for minimum necessities needed to survive). This situation is described in a passage of Laidi Fernandez de Juan's story titled "Clemency Under the Sun" as well as in the interview one of us was able to conduct with her in Havana which can be consulted in (López-Cabrales 2007a).

[5] Many female writers told one of us that despite everything, they would not want to move away from Havana. Marylin Bobes even explained to one of the authors that she lived with her ex-husband in Paris for three months and had to return, breaking her relationship with him during the special period. Rosario Cardenas, a choreographer and creator of combinatory dance, currently living in Cuba, told us that, indeed, she had received offers to stay and live in the US, Mexico, Spain, and Australia and that she did not have a very concrete response as to why she turned them down. "[...] if it is clear to me that to this moment, my internal self has rebelled. I believe that because none of those offers, however lofty in scale, would make it possible that my roots were not trampled on, and if that were to happen, how could I make my tree grow?" (López-Cabrales 2007a).

[6] Although the reconversion of the country at large was a process that occurred from the 1970s to the 1990s and that has had, as a consequence, an infinite number of social changes or movements in different industrial sectors.

[7] San Fernando, the birthplace of Ana Rossetti and where she has resided most of her life, is commonly referred to as "the island".

[8] "She gestured to the disastrous, unsavory women that peeked down from the heights of stairwells, the partners of the one that had fainted, and between the two of them, they took the body that they dragged towards the bedroom. The remaining men that had crowded around the lintel of a cupboard, where they preceded the shoe test that Andres provided them with, were ordered that she be placed on a speedboat, taken to sea, and once there, dumped to accompany so many others. No traces, or fingerprints, or calls for attention. That would be just what we needed right now" (López-Cabrales 2012).

[9] On that occasion, the imaginary architecture that was created from a distance (for example, the Nanban screen that contains the image of the city of Seville protected by mountains that helped to preserve the space, but that did not exist in reality because the city is located in the vast valley of the Guadalquivir River) induced a concept of protective location, as Feng-Shui itself recommended. Hence, the visual representations of this great city in the century following the founding of Nagasaki produced images where a sustainable idea of space predominates (Cabeza-Lainez and Rodríguez-Cunill 2019). In the cities created by the protagonist women writers of this text, we find the underlying portrait of survival in a complex world and detail in the characters' lives, and less in the city or the urbanism that sustains it.

10   The list would be endless: Pizán, Quevedo, Bocaccio, Joyce, Dickens, Scott Fitzgerald, Pamuk... Yet, we have focused only on two cities because of their literary and cultural relationship.

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
