# Peer review of "From Havana to Cádiz in the Imaginary of Women Writers of the Last Decades"

_2410-9789, doi:10.3390/literature3020017_

Round 1

Reviewer 1 Report

This might be a compelling article, but it contains some problems in its present form. Why is the topic relevant to you, or to the reader? The abstract needs to be written in simple, clear English. No clear methodology seems to be in operation. George Bataille and Diana Agrest are listed as possible theoretical backgrounds, but they do not appear in the discussion part of your essay. Some of the books mentioned, such as Ena Lucía Portela's (One Hundred Bottles, found on the Goodreads website) have been translated. It might be advisable to provide titles and quotations in the original language, and if necessary, translate titles and quotations that have not been translated,  but consistently (Night Watch/Night Round). A stronger conclusion would also improve your article.

Reviewer 2 Report

The topic is quite interesting; however, the argument and analysis requires more development. Section 3 “Havana and Cadiz, literary subjects” requires the most attention. The citations are plentiful and cumbersome, often obscuring the author’s voice and context. The essay also lacks a conclusion, and instead abruptly ends with the reference of “a Moroccan girl,” which is disconnected from the essay context. The last paragraph hurriedly mentions a litany of literary references without much analysis or context. It is not enough to identify similarities. 

Round 2

Reviewer 1 Report

I suggest you proofread your text once more before submitting it. I found "The Cotidian Nothing" (108, 220), "The Quotidian Nothing" (277), when apparently the text has been translated as "The Daily Nothing"; see: Zoé Valdés, a Pen Like a Whip / Iván García – Translating Cuba

This sequence is hard to understand: that lives braiding her long here (380); are you referring to "here" (present) metaphorically, or "hair"?

Reviewer 2 Report

This revision is a notable improvement. Please note the following:

1: “But Cadiz has also become a conflictive area due to the immigration of people from sub-Saharan and North African communities to the province.” — Terms such as “sub-Saharan African” and “North African” are deeply problematic due to their colonialist / white supremacist / racist origins. It would be equally inclusive to simply state “immigration of people from the African continent.”

2: Lines 277-288 seem to belong in a different part of Section 3. Review the organization.

3: Proofread for spelling and grammar. For example, the word “here” was used and it should probably be “hair.” Also, is it “boot for the right foot?” and not “right food”?

4: The documentation format for longer in-text citation / block quotes needs some editing, especially the use of quotation marks. Should this literary essay be in MLA format?
